# Physical Fitness and Body Composition of Youth Referees in Invasion Games

**DOI:** 10.3390/children10040650

**Published:** 2023-03-30

**Authors:** Élvio Rúbio Gouveia, Helder Lopes, Marcelo de Maio Nascimento, Filipe Manuel Clemente, Hugo Sarmento, Andreas Ihle, Gerson Ferrari, Pedro Luz, Henrique Caldeira, Adilson Marques

**Affiliations:** 1Laboratory of Robotics and Engineering Systems (LARSYS), Interactive Technologies Institute (ITI), 9020-105 Funchal, Portugal; 2Department of Physical Education and Sport, University of Madeira, 9020-105 Funchal, Portugal; 3Research Center in Sports Sciences, Health Sciences, and Human Development (CIDESD), 5001-801 Vila Real, Portugal; 4Center for the Interdisciplinary Study of Gerontology and Vulnerability, University of Geneva, 1205 Geneva, Switzerland; 5Department of Physical Education, Federal University of Vale do São Francisco, Petrolina 56304-917, Brazil; 6Escola Superior Desporto e Lazer, Instituto Politécnico de Viana do Castelo, Rua Escola Industrial e Comercial de Nun’Álvares, 4900-347 Viana do Castelo, Portugal; 7Instituto de Telecomunicações, Delegação da Covilhã, 1049-001 Lisboa, Portugal; 8University of Coimbra, Research Unit for Sport and Physical Activity (CIDAF), Faculty of Sport Sciences and Physical Education, 3004-504 Coimbra, Portugal; 9Department of Psychology, University of Geneva, 1205 Geneva, Switzerland; 10Swiss National Centre of Competence in Research LIVES—Overcoming Vulnerability: Life Course Perspectives, 1015 Lausanne, Switzerland; 11Escuela de Ciencias de la Actividad Física, el Deporte y la Salud, Universidad de Santiago de Chile (USACH), Santiago 9170022, Chile; 12Facultad de Ciencias de la Salud, Universidad Autónoma de Chile, Providencia 7500912, Chile; 13Interdisciplinary Centre for the study of human Performance (CIPER), Faculty of Human Kinetics, University of Lisbon, 1495-751 Lisbon, Portugal; 14Instituto de Saúde Ambiental (ISAMB), Faculty of Medicine, University of Lisbon, 1649-020 Lisbon, Portugal

**Keywords:** agility, body fatness, cardiorespiratory fitness, children and adolescents, collective sports, physical education, strength, sports practice, referees’ activity

## Abstract

Background: Referees’ activity can be an instrument for training students with an impact on health-related physical fitness (PF). This study aimed to investigate the differences in PF and body composition between students without sports practice (G1), students with regular sports practice (G2), and student referees in team invasion games (G3). Methods: This study followed a cross-sectional design. The sample comprised 45 male students between 14 and 20 years old (16.40 ± 1.85). Three groups (G1, G2, and G3) of 15 participants each were selected. PF was assessed by a 20 m shuttle run, change-of-direction test, and standing long jump. Body composition was determined by body mass index (BMI; kg/m^2^) and the percentage of body fat (%BF) predicted by skinfold. Results: After controlling for age as a covariate, the block of variables used to characterize PF was statistically different between sports practice groups favoring student referees (*p* < 0.001; r = 0.26). Similar results were achieved for body composition (i.e., BMI and %BF) (*p* < 0.001; r = 0.17). However, when the dependent variables were evaluated separately, there were only differences between groups in %BF (*p* = 0.007; r = 0.21). Student referees had statistically significantly lower values than the remaining groups. Conclusion: Refereeing activity benefits PF related to health and performance, including body composition. This study confirms the benefits related to health in children and adolescents who are involved in refereeing activity.

## 1. Introduction 

Referees and players must have a high physical fitness (PF) [1]. During a sporting match, the referee must remain as close as possible to the game situation, controlling the players’ actions so that the match between the teams follows the rules. Thus, if the referee’s PF is low, adverse events may occur, impairing the progress of the match as well as the psychological state of the players [2].

During the match, a referee, besides having an excellent PF, must be able to make quick decisions, which is essential to solving disagreements and maintaining the order of the game [3]. Referees work in high-pressure environments, which requires high self-efficacy [4], which is understood as a condition for successfully solving different levels associated with a task [5]. Individuals with high self-efficacy potential can concentrate, change their action strategy quickly, and recover from stressful situations without attributing faults to themselves [6]. Altogether, we hypothesize that referees may have a better general health profile (based on PF) than people who do not practice sports.

In a study carried out by Adé et al. [7], the authors evaluated physical education (PE) classes regarding the social roles of mentor and coach for 74 students (39 girls; M = 13.54 years, SD = 2.09). The purpose was to investigate whether mentoring and coaching would benefit motor, methodological, and social learning. The study brought to light unknown information about the role of the student referee in PE classes. The principle suggested by the authors was that when the student assumes the role of referee, he automatically experiences another facet of the sport called the experience of responsibility. Furthermore, this is what PE teachers seek to develop with students during classes. Based on this principle, to our knowledge, no study has evaluated whether PE students who perform referee activity present differences in PF and body composition compared to other students who practice and do not practice sports. Thus, the experience of responsibility mediated by sport may increase the perception of the importance of achieving and maintaining adequate health and physical well-being.

Previous studies have shown that due to the physical demands of sports matches, referees tend to be engaged to reach higher levels of PF [8,9]. In invasion collective sports such as football, basketball, or handball, referees are expected to have adequate aerobic power [10]. It is worth noting that in situations of high physical exertion, individuals with low PA may have cognitive function deficits, impairing decision making in competitive situations [11]. Moreover, referees with low physical conditioning are likelier to be more tired during the match, increasing the chances of injuries [12]. Another point to consider about the possible benefits of involving PE students with refereeing to maintain adequate PF levels (i.e., strength, aerobic capacity, agility) are the requirements of national and international refereeing regulatory institutions (i.e., UEFA and FIFA) for professional practice [13]. Thus, referees are regularly evaluated physically and must reach pre-established PF levels [14]. According to Casajus and Castagna [10], professional soccer referees remain physically active for a long time, reaching their best performance at 40. It is usual for players to have already ended their careers in this age group.

Previous studies have shown that better PF levels were positively associated with reduced body fat [15]. PA is a predictor of morbidity and mortality from chronic diseases [16,17]. Thus, referee PE students may develop superior healthcare behavior compared to students who are not used to the challenging situations of responsibility, judgment, and decision making that referees face.

Most studies on sports referees have focused on their PF [18,19,20], which in turn impairs the referee’s performance during games and also affects emotional dimensions and decision making [4,21,22]. Moreover, concerning PE students, the literature is vast on the level of PF and body composition [23,24,25]. However, in the literature, there is no specific information about the PF levels and body composition of the referee PE student. Thus, this study aimed to investigate PF and body composition differences between students without sports practice, students with regular sports practice, and PE student referees in collective games of invasion sports.

## 2. Materials and Methods

### 2.1. Study Design and Ethical Approval

This study followed a cross-sectional design aiming to compare three groups of sub-populations. A non-probabilistic convenience sampling was conducted among youth students from middle and high schools. The inclusion criteria were: group 1: (1) belonging to the Project “Physical Education in Schools from the Autonomous Region of Madeira” (EFERAM-CIT) and (2) aged between 14 and 20 years old, not practicing any sport or regular physical activity weekly; group 2: (1) belonging to the EFERAM-CIT, (2) aged between 14 and 20 years old, having a regular practice of any sport of exercise ≥7 months per year and with a weekly load of ≥3 h; group 3: (1) being registered at the regional soccer federation as a referee at least 6 months before the assessments and (2) being between 14 and 20 years old. This study was approved by the Regional Secretary of Education, the school’s headmaster, and the Scientific Committee of the Faculty of Physical Education and Sports at the University of Madeira (Reference: ACTA N.77-12.04.2016). All the participants and their legal guardians were informed about the study’s objectives, and written informed consent was obtained.

### 2.2. Settings and Context

The study lasted 25 days, and it was undertaken between October 2022 and January 2023. This corresponds to the academic year’s duration. The data collection occurred during the PE sessions for group 1 and group 2. Group 3 was assessed in the same conditions but at the University of Madeira. Graduates in PE and Sport conducted the assessments. The field team underwent a training and preparation phase for the evaluations before the assessments. In this training, the instructions for carrying out the PF and body composite tests were discussed and practiced. This training phase culminated in a pilot study in eight boys and seven girls aged 16–18 years. All participants were assessed twice, with one week between assessments. We verified high reliability in the assessments, with values of the interclass correlation coefficient between 0.797 and 0.999.

### 2.3. Participants

The study included 45 male students between 14 and 20 years old (16.40 ± 1.85) who were middle and high school students. The participants of this study are part of the research project EFERAM-CIT (https://eferamcit.wixsite.com/eferamcit, accessed on 15 February 2023). Three groups of 15 participants each were selected: group 1: students without sports practice; group 2: students involved in regular sports practice; group 3: student referees.

### 2.4. Methodological Procedures

The assessments were conducted on the same day. The sequence of tests was as follows: (i) anthropometry; (ii) standing long jump; (iii) change-of-direction test; and (iv) 20 m shuttle run. A 5 min rest was implemented between tests. After the anthropometry, the participants were subjected to a standardized warm-up protocol consisting of general aerobic exercises, mobility, and flexibility. After the warm-up, the participants performed one trial of the standing long jump interspaced by 1 min rest. In the next test, the participants performed one trial of a 4 × 10 m change-of-direction test interspaced by 1 min. Finally, they performed the 20 m shuttle run test.

### 2.5. Anthropometry and Body Composition

Body mass, stature, and adiposity skinfolds (triceps and calf) were evaluated according to the procedures described in the FitnessGram test battery [26]. Body mass (kg) and standing height (mm) were measured using a scale and a SECA stadiometer (Model 761, Hamburg, Germany, and Model 213, Hamburg, Germany, respectively). Then, body mass index was calculated based on those metrics (kg/m^−2^). The Skinfold Caliper (Harpenden, UK) was used to assess the skinfolds. The triceps and calf skinfolds were used to calculate the % body fat [27].

### 2.6. Standing Long Jump

The participants were instructed to jump from the starting line as far as possible. The distance between the starting line and the heel determined the standing jump score. Two measures were performed, and the best score was considered in the analysis.

### 2.7. 20 m Shuttle Run

The progressive aerobic cardiovascular endurance run (PACER) test from Fitnessgram was assessed as an indicator of cardiorespiratory fitness. The shuttle test consists of executing the maximum number of routes over a distance of 20 m at a predetermined cadence. A detailed description of the evaluation procedures, equipment, scoring, and safety precautions can be found in the Fitnessgram/Activitygram Test Administration Manual [26].

### 2.8. Data Analysis

First, a descriptive statistics analysis by means and standard deviation was calculated. In this first phase, preliminary assumption testing was conducted to check for normality, linearity, univariate and multivariate outliers, homogeneity of variance-covariance matrices, and multicollinearity.

Second, a one-way between-groups analysis of variance was conducted to explore the differences in age, PF tests, and body composition variables between three groups: group 1: students without sports practice; group 2: students involved in regular sports practice; group 3: student referees. Additionally, multiple comparisons considering the Bonferroni test were performed.

Finally, a one-way between-groups multivariate analysis of variance was performed to investigate students’ different profiles (i.e., G1: students without sports practice; G2: students involved in regular sports practice; G3: student referees) in PF tests and body composition. All the analyses were performed considering age as a covariate. Data analysis was performed using IBM SPSS v29 (IBM Corp., Armonk, NY, USA). The significance level was set at *p* < 0.05.

## 3. Results

Table 1 depicts descriptive statistics on age, physical fitness tests, and body composition according to different groups. Descriptive statistics on age, physical fitness tests, and body composition according to different groups (students without sports practice—G1; students involved in regular sports practice—G2; student referees—G3). There was a statistically significant age difference (F (2, 44) = 6.7; *p* = 0.003), SR (F (2, 44) = 27.8; *p* < 0.001), SLJ (F (2, 44) = 9.5; *p* < 0.001), change-of-direction test (F (2, 44) = 13,0; *p* < 0.001), and %BF (F (2, 44) = 10.3; *p* < 0.001) for the three groups. Post hoc comparisons using the Bonferroni test indicated that the group of student referees was significantly different from students without sports practice in SR, SLJ, change-of-direction, and %BF and from the group with regular sports practice in SR and %BF. In both cases, the referees group presented better scores in the physical tests and lower %BF. Significantly higher scores in the PF tests (i.e., SR and change-of-direction) were seen in the group with regular sports practice compared to those without sports practice in SR and change-of-direction test.

The results of the study revealed that in PF, three dependent variables were used: the number of laps in a 20 m shuttle run (Figure 1), change-of-direction test (sec) (Figure 2), and standing long jump (Figure 3). The independent variable was students’ profiles (i.e., students without sports practice; students involved in regular sports practice; student referees). After considering age as a covariate, there was a statistically significant difference between students on the combined dependent variables: F (6, 80) = 4.78, *p* < 0.001; Pillai’s Trace = 0.53; partial eta squared = 0.26.

When the results for the dependent variables were considered separately, all dependent variables reached statistical significance using a Bonferroni adjusted alpha level of 0.017, ps < 0.010; 0.20 < partial eta squared > 0.44. An inspection of the mean scores indicated that student referees reported slightly better scores on the number of laps in the 20 m shuttle run, change-of-direction test (sec), and standing long jump.

Regarding body composition, two dependent variables were used: %BF and BMI. There was a statistically significant difference between students on the combined dependent variables: F (4, 82) = 4.26, *p* < 0.001; Pillai’s Trace = 0.34; partial eta squared = 0.17.

When the results for the dependent variables were considered separately, the only difference to reach statistical significance was the %BF, F (2, 45) = 5.53, *p* = 0.007, partial eta squared = 0.21. An inspection of the mean scores indicated that student referees reported a slightly lower of %BF than students without sports practice or even students involved in regular sports practice (Figure 4).

## 4. Discussion

This study aimed to investigate PF and body composition differences between students without sports practice, students with regular sports practice, and student referees in team invasion sports games. The results showed that after controlling for age as a covariate, the block of variables used to characterize PF (i.e., 20 m shuttle run, change-of-direction test, and standing long jump) was statistically different between sports practice groups favoring the student referees. Similar results were achieved for body composition (i.e., BMI and %BF). However, when the dependent variables were evaluated separately, there were only differences between groups in %BF. Student referees had statistically significantly lower values than the remaining groups.

In this study, we used the 20 m shuttle run test as an indicator of cardiorespiratory endurance. Cardiorespiratory endurance is a health-related component of PF because a positive correlation has been demonstrated between cardiorespiratory endurance and a healthier health profile [28]. Some authors even refer to cardiorespiratory endurance as a measure of total body health [29,30]. Although the most recent investigation has shown that the 20 m shuttle run test is not a valid measure to predict the maximum oxygen consumption (VO2 max), especially in children and adolescents [31,32], this has been a widely recommended test in the literature [26,30]. However, our study considered the number of laps performed in the test as an indicator of cardiorespiratory endurance, as shown previously [33,34]. In this study, young referee students showed better results in the 20 m shuttle run test than peers with and without regular sports practice, reflecting a good cardiorespiratory capacity and a healthier health profile.

The standing long jump is one of the most-used tests in the evaluation of explosive strength since it is considered a fundamental skill in a variety of sports that require complex motor coordination of the legs and arms and where a high velocity of muscle contractions is required [35]. Standing long jump is an indicator of power, the ability, or rate at which one can perform work [36], directly impacting performance-related PF. Evidence shows that explosive strength assessed with different jumps has been considered an important predictor of speed and change of direction [37,38]. However, the relationship between the standing long jump and the maximum isokinetic strength of the lower limbs is still under debate [39]. However, recent studies have considered and recommended the standing long jump as an important test index for middle school students to assess their PF levels [35]. Similar to the cardiorespiratory results achieved, the young student referees had better scores in the standing long jump performance compared to their colleagues with and without regular sports practice, which shows better muscular performance and therefore a more robust PF profile.

Another performance-PF parameter analyzed in this study was assessed with the change-of-direction test, defined as the ability to change the body’s position in space with speed and accuracy [36]. Change of direction, an indicator of agility, is considered a central component of sports training due to its beneficial impact on mental and PF [40]. For example, some studies using different population profiles found that physical agility training is likely more effective than linear running considering physical and cognitive performance such as physical agility, memory, and vigilance [40,41,42]. These results align with those obtained for cardiorespiratory endurance and explosive strength. Overall, this study supports the idea that young student referees have a healthier PF profile compared to peers who do not exercise or even those who regularly practice physical activity. These results help clarify that a referee’s activity in collective invasion sports games, even at a young age, requires high levels of physical demand to follow the game and make decisions correctly, as established in the elite professional referees’ context [43]. The involvement of younger people in the refereeing career has added advantages from the point of view of PF related to health and performance and should be used as a strong argument for attracting more referees to training.

In line with previous results in PF tests concerning body composition, statistically, significant differences were identified in the %BF between student referees (13.0%), students without sports practice (24.9%), and students with regular sports practice (21.5%). In the case of BMI, no differences were found between groups. These results support two fundamental ideas. First, body composition, a health-related PF component that expresses the relative amount of muscle, fat, bone, and other vital body parts [36], presents a healthier profile in student referees. This should be considered a great health advantage for this group. Currently, and mainly because of the COVID-19 pandemic, much concern has been expressed about overweight and obesity by the leading public health organizations in the world (i.e., WHO, Centers for Disease Control and Prevention, American Psychological Association, World Obesity Federation, and Canadian Task Force on Preventive Health Care, among others). It seems clear that being overweight and obese represents a severe problem affecting many children and adolescents across the world [44]. Today, the direct relationship between overweight and obesity and several chronic diseases including hypertension, metabolic syndrome, type 2 diabetes mellitus, stroke, cardiovascular disease, and dyslipidemia is unequivocal [36,44].

Secondly, our study draws attention to body-composition-assessment methodologies, particularly in children and adolescents of these ages (±16.4 years old). Although there are recommendations for using BMI in the definition of overweight and obesity in children and adolescents [44], its exclusive use can be problematic regardless of the cut-off values used. Our results support this argument, especially when comparing children and adolescents with and without sports practice. Body composition assessment using subcutaneous adiposity skinfolds to estimate %BF has been well correlated with several measurements considered the gold standard in the assessment of body composition [36]. Assuming the assessment technique is correctly performed, using adiposity skinfolds to estimate %BF is preferable at these ages [36].

This study addresses an important topic that has been little studied, particularly in children and young referees, and that can help to promote the benefits of refereeing in collective sports games at these ages. However, the interpretation of the results of this study must be made considering some limitations. First, the sample is small. However, considering the population of student referees with inclusion criteria, we assessed 60% of the population in the region where the study was conducted. Furthermore, we acknowledge that to avoid sample imbalances, the constitution of the remaining groups (i.e., without sports practice and with regular sports practice) was conditioned. We also acknowledge that the PF and body composition metrics used in this study depend on the technician’s expertise. This means that proper training and sample practice of the technique is necessary to obtain accurate measurements. However, to deal with this, the data were collected by well-trained team members who graduated in PE and sports, ensuring protocol consistency, and minimizing data-collection errors, especially in anthropometric measures. Additionally, the statistical analyses considered the control of important covariates that could create bias in the results.

## 5. Conclusions

Information about the PF and body composition of youth referees in invasion games is scarce in the literature. We conclude in this study that after controlling for age as a covariate, student referees in collective games of invasion sports presented advantages in PF and body composition when compared with students without sports practice or even with students with regular sports practice. This study draws attention to the importance of using cross-measurements in assessing body composition in children and adolescents, as is the case with anthropometric measurements to predict %BF. Finally, this study confirms the benefits related to health in children and adolescents who opt for the refereeing activity.

## Figures and Tables

**Figure 1 children-10-00650-f001:**
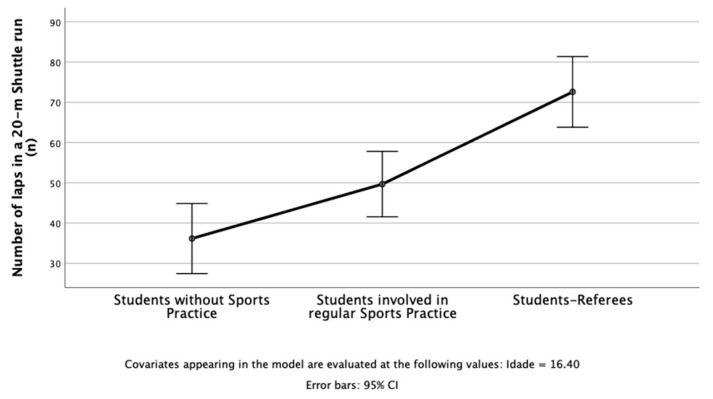
Students without sports practice, students involved in regular sports practice, and student referees in the number of laps in a 20 m shuttle run.

**Figure 2 children-10-00650-f002:**
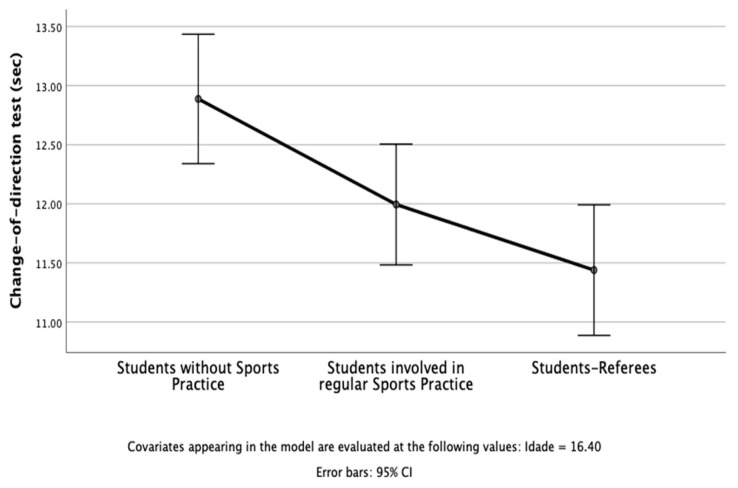
Students without sports practice, students involved in regular sports practice, and student referees in the change-of-direction test (seconds).

**Figure 3 children-10-00650-f003:**
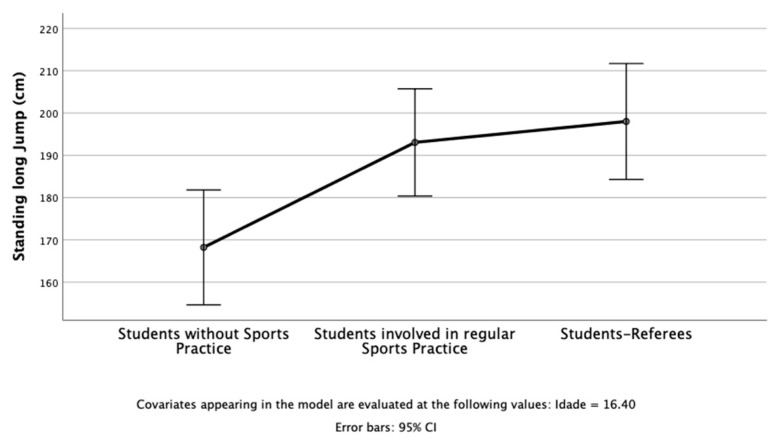
Students without sports practice, students involved in regular sports practice, and student referees in the standing long jump (cm).

**Figure 4 children-10-00650-f004:**
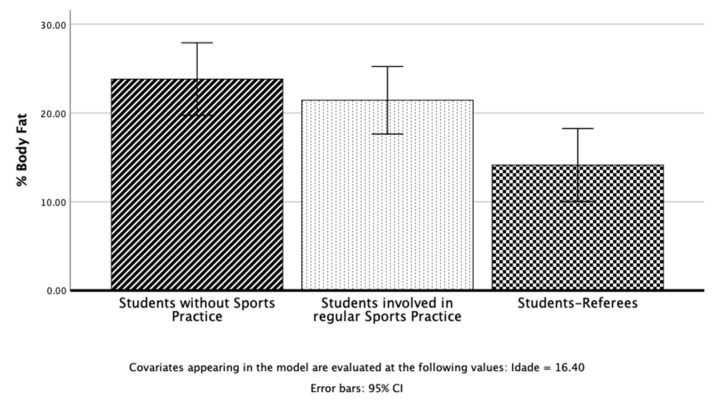
Students without sports practice, students involved in regular sports practice, and student referees in the %BF.

**Table 1 children-10-00650-t001:** Descriptive statistics on age, physical fitness tests, and body composition according to different groups (students without sports practice—G1; students involved in regular sports practice—G2; student referees—G3).

	Total		S-w/SP(G1)		S-reg/SP(G2)		S/REF(G3)			
	M	SD	M	SD	M	SD	M	SD	‡	†
Age (years)	16.4	1.9	15.3	1.5	16.3	1.5	17.5	1.9	*p* = 0.003	1 < 3
SR (n)	52.8	25.0	31.9	16.9	49.4	19.7	77.1	13.0	*p* < 0.001	1 < 2 and 3, 2 < 3
SLJ (cm)	186.4	29.2	164.3	30.2	192.8	28.2	202.2	11.7	*p* < 0.001	1 < 3
Chan-D (sec)	12.1	1.3	13.1	1.4	12.0	0.8	11.2	0.6	*p* < 0.001	1 < 2 and 3
BMI (kg/m^2^)	21.8	2.9	21.6	2.8	21.7	3.7	22.0	2.3	*p* < 0.913	n.s.
%BF	19.8	8.8	24.9	8.5	21.5	8.6	13.0	4.2	*p* < 0.001	1 and 2 > 3

BMI, body max index; M, mean; SD, standard deviation; SLJ, standing long jump (cm); Chan-D, change-of-direction; SR, 20 m shuttle run (n); S-w/SP(G 1), students without sports practice; S-reg/SP(G2), students involved in regular sports practice; S/REF(G3), student referees; %BF, percentage of body fatness estimated from skinfolds; ‡, one-way ANOVA; †, multiple comparisons considering Bonferroni correction.

## Data Availability

The data presented in this study are available on request from the corresponding author.

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
