# Peer review of "Physical Fitness and Body Composition of Youth Referees in Invasion Games"

_children, 2023, doi:10.3390/children10040650_

Round 1

Reviewer 1 Report

Nowadays, invasion sports such as football have acquired very high levels of physical, technical and tactical preparation. Therefore, the performance of referees must be at the same level as that of the players. Hence the importance of studying the physical demands of referees in sporting competition.   The conclusion of the study does not correspond to the stated objective; it should strictly respond to this objective. It talks about investigating the differences in physical preparation and body composition, but in the conclusion it states that it is more beneficial to carry out refereeing activities, when this is not directly related to the object of the study.   On the other hand, it would be convenient in future studies where anthropometric measurements are taken to use a Body Composition Analyser, a machine that uses Electrical Bioimpedance to evaluate and quantify in each person the different parameters, essential to correct, adapt and personalise a training plan.

Author Response

Physical Fitness and Body composition of Youth Referees in Invasion Games

Thank you for the opportunity to revise our manuscript and resubmit it to Children. We are very grateful to the reviewers for their overall positive evaluation and their helpful suggestions for improvement. In a thorough revision, we have addressed all the comments raised by the reviewers, and we believe that the manuscript has substantially improved.

We hope the revised version is now suitable for publication in Children.

The changes we made in the article are shown using track changes.

Yours sincerely,

Élvio Rúbio Gouveia

Responses:

1 - The conclusion of the study does not correspond to the stated objective; it should strictly respond to this objective. It talks about investigating the differences in physical preparation and body composition, but in the conclusion, it states that it is more beneficial to carry out refereeing activities, when this is not directly related to the object of the study.  

Response: We thank the reviewer's comment. We revised the conclusions, to agree with the objectives.

2 - On the other hand, it would be convenient in future studies where anthropometric measurements are taken to use a Body Composition Analyser, a machine that uses Electrical Bioimpedance to evaluate and quantify in each person the different parameters, essential to correct, adapt and personalise a training plan.

Response: The reviewer is right in his suggestion. Typically, in children, it might even be better to have a combination of body composition assessment instruments. Unfortunately, in this study, we do not have information on Electrical Bioimpedance. Our choice was anthropometric measurements, because it is more economical, and because it is recommended as an alternative and reliable field measurement. To eliminate assessment errors, the evaluation team was trained, and a pilot study was carried out to assess the reliability of the evaluations. In all analyzed metrics, the infraclass correlation coefficient, considering two assessments of the same individual a week apart, were all above 0.70.

Reviewer 2 Report

Dear Editor and Authors,

Thank you for the opportunity to review this manuscript, which is very interesting and current as it aims to compare differences in physical fitness and body composition among athletes between inactive students who practice regular physical activity and student referees who play in invasion games.  Certainly a limitation is the sample size, as described by the authors in the discussion. However, it is an initial study that can be continued by expanding the sample.

The study found statistically significant differences in favor of student referees, emphasizing the importance of referee activity in improving health and performance, which are currently crucial parameters to work on given the poor levels of movement of this group and others globally.  I have some suggestions that can help improve the manuscript, including a revision of the English language to ensure linearity in reading. Keywords should include words not already in the title.  Some English words should be revised for example line 69 and 72 line 150 line 151 (aerobibic exercises, bomility) and 153. Also the word physical education previously abbreviated to PE in the method is repeated several times without the abbreviation so I recommend correcting it.  On the other hand, regarding the statistical analysis, the author first states that he used descriptive statistics then One-Way ANOVA, after which he talks about preliminary assumptions to check normality, linearity etc..which should come before the of the ANOVA calculation so I would suggest putting that ruling second, and then talk about One-way ANOVA, Bonferroni and the multivariate.

In the results I would avoid rewriting that "A one-way between-groups analysis of variance was conducted to explore the impact ..." rather I would rephrase the sentence by specifying that the results of the study revealed that... The same applies to the other results. 

In the discussion more attention should be paid to the way the citations were inserted as there are some citations left with APA style e.g. line 280.  All tenses should be in past tense and not present tense.

Author Response

Physical Fitness and Body composition of Youth Referees in Invasion Games

Thank you for the opportunity to revise our manuscript and resubmit it to Children. We are very grateful to the reviewers for their overall positive evaluation and their helpful suggestions for improvement. In a thorough revision, we have addressed all the comments raised by the reviewers, and we believe that the manuscript has substantially improved.

We hope the revised version is now suitable for publication in Children.

The changes we made in the article are shown using track changes.

Yours sincerely,

Élvio Rúbio Gouveia

Responses: 

1- The study found statistically significant differences in favor of student referees, emphasizing the importance of referee activity in improving health and performance, which are currently crucial parameters to work on given the poor levels of movement of this group and others globally.  I have some suggestions that can help improve the manuscript, including a revision of the English language to ensure linearity in reading. Keywords should include words not already in the title.  Some English words should be revised for example line 69 and 72 line 150 line 151 (aerobibic exercises, bomility) and 153. Also the word physical education previously abbreviated to PE in the method is repeated several times without the abbreviation so I recommend correcting it.  

Response: We appreciate the reviewer's positive comments on the article. A more careful revision of the English was carried out throughout the text.

2 - On the other hand, regarding the statistical analysis, the author first states that he used descriptive statistics then One-Way ANOVA, after which he talks about preliminary assumptions to check normality, linearity etc.. which should come before the of the ANOVA calculation so I would suggest putting that ruling second, and then talk about One-way ANOVA, Bonferroni and the multivariate.

Response: We agree with the reviewer's suggestions and have changed the text in order to be more aligned with the steps that were carried out throughout the analyses.

3 - In the results I would avoid rewriting that "A one-way between-groups analysis of variance was conducted to explore the impact ..." rather I would rephrase the sentence by specifying that the results of the study revealed that... The same applies to the other results. 

Response: We agree with the reviewer's suggestion. To avoid duplication of information, we removed the information about the statistical analysis from the results.

4 - In the discussion more attention should be paid to the way the citations were inserted as there are some citations left with APA style e.g. line 280.  All tenses should be in past tense and not present tense.

Response: We appreciate the reviewer's comment. A text check was carried out to correct these errors.